# Development and evaluation of an online HIV pre-exposure prophylaxis (PrEP) training program for community pharmacists to implement pharmacy-led PrEP services in Malaysia

Yan Nee Gan[1,2], Rafdzah Zaki[1,3]*, Safia Alia Hafidzar[4], Kwee Choy Koh[5],
Mariani Ahmad Nizaruddin[6], Rosnida Mohd Noh[7], Khairil Erwan Khalid[8], Andrew Yap[9],
Frederick L. Altice[4,10,11,12], Sin How Lim[1,4], Iskandar Azwa[4,13]

**1** Department of Social and Preventive Medicine, Faculty of Medicine, Universiti Malaya, Kuala Lumpur, Malaysia, **2** Malaysian Health Technology Assessment Section, Ministry of Health, Putrajaya, Malaysia, **3** Centre for Epidemiology and Evidence-based Practice (CEBP), Universiti Malaya, Kuala Lumpur, Malaysia, **4** Centre of Excellence for Research in Infectious Diseases and AIDS (CERiA), Universiti Malaya, Kuala Lumpur, Malaysia, **5** Department of Medicine, IMU University Clinical Campus, Seremban, Negeri Sembilan, Malaysia, **6** Department of Community and Pharmacy Practice, Faculty of Pharmacy, University of Cyberjaya, Cyberjaya, Selangor, Malaysia, **7** Department of Medicine, Faculty of Medicine, Universiti Teknologi MARA, Sungai Buloh, Selangor, Malaysia, **8** Department of Medicine, Hospital Kuala Lumpur, Kuala Lumpur, Malaysia, **9** The Red Clinic, Petaling Jaya, Selangor, Malaysia, **10** Section of Infectious Diseases, Department of Internal Medicine, Yale School of Medicine, New Haven, Connecticut, United States of America, **11** Department of Epidemiology of Microbial Diseases, Yale School of Public Health, New Haven, Connecticut, United States of America, **12** Center for Interdisciplinary Research on AIDS (CIRA), Yale University, New Haven, Connecticut, United States of America, **13** Infectious Diseases Unit, Department of Medicine, Faculty of Medicine, Universiti Malaya, Kuala Lumpur, Malaysia

* rafdzah@ummc.edu.my

## Abstract

### Introduction

Expanding HIV pre-exposure prophylaxis (PrEP) through pharmacies may improve access for key populations. As part of the preparation phase of the EPIS (Exploration-Preparation-Implementation-Sustainment) framework, we developed and evaluated an online, self-paced PrEP training program for community pharmacists to prepare for a pilot, pharmacy-led PrEP service in Malaysia.

### Materials and methods

In May 2023, a PrEP training committee developed an online training program covering PrEP efficacy, safety, eligibility, baseline assessments, laboratory testing, prescribing, and special circumstances, and pre-/post-training knowledge tests. In June 2023, 18 community pharmacists asynchronously completed the training. Effectiveness was evaluated using a 20-question pre-/post-training knowledge test, with responses calculated into percentage scores, alongside participant feedback. Paired t-tests assessed knowledge score differences ($p < 0.05$).

---

**Data availability statement:** The relevant data are available in the Harvard Dataverse at https://doi.org/10.7910/DVN/BONKBK.

**Funding:** This research was funded by the Malaysian Implementation Science Training program (MIST) for YG supported by Fogarty International Center, NIDA, NIMH and NICHD (D43 TW-011324 – PI: Frederick L. Altice, Adeeba Kamarulzaman) https://www.fic.nih.gov/ and the World Health Organization (WHO) (WHO registration: 2023/1363112-0) for IA https://www.who.int/malaysia. The funders had no role in study design, data collection and analysis, decision to publish, or preparation of the manuscript.

**Competing interests:** The authors have declared that no competing interests exist.

## Results

Participants (median age: 30.5 years [IQR: 5.8]; 78% female; 89% Chinese; median 6.5 years of experience [IQR: 4.5]; four with prior HIV-related experience) showed a mean increase in knowledge scores of 14.2% (95% CI: 8.2%–20.1%; $p < 0.001$), increasing from 64.7% to 78.9%. Scores for four mid-career participants (50% female, 5–10 years of experience, all with undergraduate degrees, including one with prior HIV-related training), however, did not improve, suggesting that tailored learning approaches may be needed, and that existing knowledge or prior experience do not necessarily predict learning outcomes. PrEP knowledge gaps remained primarily in counseling (−22%), identifying candidates (−12%), clinical contraindications (−6%), effectiveness (−6%), and management of missed doses for daily PrEP (−6%), highlighting potential challenges in clinical decision-making and patient communication. Most pharmacists agreed that the training was well-structured, easy to understand, of appropriate duration, and useful for their work. Satisfaction was high, as was willingness to recommend it.

## Conclusions

The novel online self-paced training program improved pharmacists' PrEP knowledge, though variability in knowledge gains suggests the need for enhancements. Incorporating case-based, problem-based, and simulation-based learning may improve comprehension, particularly in patient counseling, eligibility assessment, and PrEP contraindications.

## Introduction

In Malaysia, the HIV epidemic remains concentrated in key populations such as men who have sex with men (MSM), people who inject drugs (PWID) and transgender women, each with elevated HIV prevalence of 12.9%, 7.5%, and 5.9%, respectively [1]. Despite national efforts, annual HIV infections continue to rise over the past decade, driven primarily by sexual transmission and hindered by suboptimal treatment and prevention coverage [2,3]. The shift from injection-related to sexual transmission underscores the urgent need to scale up HIV pre-exposure prophylaxis (PrEP), a proven biomedical intervention for preventing sexual HIV acquisition [3].

In a setting where same-sex behavior, drug use, and transgender identities are criminalized, however, conventional models of PrEP delivery face significant structural and societal barriers [4]. Following the World Health Organization (WHO)'s 2014 recommendation, Malaysia incorporated PrEP into its National Strategic Plan (NSP) and antiretroviral therapy consensus guidelines in 2018 [5,6]. Yet, access to PrEP remains limited, primarily restricted to selected private clinics and university hospitals within the Klang Valley [7], and only recently, since January 2023, has it become available in 31 government-run primary care clinics nationwide [3].

With an estimated 220,000 MSM who could benefit from PrEP in Malaysia [8,9], current service delivery models have reached fewer than 5,000 individuals [3], highlighting the need for alternative, differentiated service delivery models beyond traditional medical facilities to expand PrEP access. The WHO recommends tailoring PrEP services to the needs and preferences of key populations to improve uptake, adherence, and overall effectiveness [10]. Community pharmacies offer a promising and innovative opportunity for expanding PrEP access through reduced stigma reinforcement and low-barrier service delivery. In Malaysia, over 3,000 community pharmacies operate with extended hours and frequent public interactions, averaging 31 visits per adult annually [11,12].

A qualitative study observed that MSM in Malaysia preferred PrEP delivery in pharmacies over current venues [13]. Furthermore, community pharmacists are well-trained healthcare professionals and well-positioned to provide PrEP service through strategic task-shifting, which involves the appropriate reallocation of responsibilities such as counselling and dispensing from physicians to pharmacists to improve service delivery and access. Globally, pharmacy-led PrEP models have been successfully implemented in settings like the United States and Kenya, with evidence supporting their feasibility and acceptability [14–18]. In Malaysia, such services remain unavailable, despite their potential to overcome barriers associated with public healthcare facilities, including stigma, overcrowding, long wait times, and limited focus on preventive care [18]. One of the main barriers to implementing PrEP services in pharmacies, however, is the lack of pharmacist training [19–21], as pharmacists in the public or private sector have been minimally exposed to this effective treatment. Studies consistently report low levels of PrEP knowledge among pharmacists [19,22–24], and preferences for receiving training through online platforms [25]. Although training and education are critical strategies for implementing evidence-based practices (EBPs), pharmacist-specific PrEP education has not yet been developed or tested.

To address this implementation gap, our study developed and evaluated an online training module tailored for Malaysian community pharmacists to prepare them for tenofovir-based oral PrEP service delivery. This training was intended to equip pharmacists with the necessary knowledge and confidence to conduct PrEP assessments and initiate telemedicine-enabled prescriptions in a future pharmacy-led PrEP model. Adequate training is essential for the successful implementation of future pharmacy-led PrEP services, as sufficient PrEP knowledge enhances pharmacists' dispensing practices and counseling confidence [24,26]. While the Malaysian Society for HIV Medicine (MASHM) has developed a self-paced online course on PrEP and post-exposure prophylaxis (PEP) for general practitioners [27], there are currently no equivalent modules tailored for pharmacists. Our newly created training strategy fills this gap by focusing on the unique roles and responsibilities of pharmacists in providing PrEP within the community pharmacy setting.

## Materials and methods

### Study design

This pre-implementation study was guided by the Exploration-Preparation-Implementation-Sustainment (EPIS) framework, focusing on the Preparation phase [28]. It involved the development (May 2023) and delivery (June 2023) of an online, asynchronous, self-paced PrEP training program for community pharmacists in the Klang Valley. Before developing the training module, a stakeholder consultation was conducted in April 2023 to identify training needs and pharmacist competencies, and to select six private community pharmacies as study sites for future PrEP service implementation. The stakeholder consultation meeting involved 31 participants, including representatives from community pharmacies, professional societies, non-governmental organizations (NGOs), telemedicine providers, and HIV researchers.

Key training needs identified by stakeholders included foundational knowledge on HIV and PrEP, PrEP eligibility and continuation criteria, and risk behavior assessment. Stakeholders also emphasized essential competencies such as person-centered communication, professional conduct, and the ability to manage client confidentiality and address stigma-related concerns. To address these different domains, knowledge-based content was delivered through online self-paced training, which is the primary focus of this manuscript. Communication skills, professional behavior, and

scenario-based role-playing sessions were covered in a supplementary in-person training workshop conducted after the completion of the online component. This workshop also included guidance on effective communication, emotional support, encouraging confirmatory testing, and collaboration with physicians and NGOs for participant follow-up and seamless referral to care.

### Training program development

A PrEP training committee comprising five experts—two from the MASHM, one from the Malaysian Pharmacists Society (MPS), and two academics—oversaw the training development. The online training program covered the fundamentals of PrEP which included efficacy, safety, eligibility criteria, initial assessment, HIV testing and other baseline tests, prescribing PrEP, and handling special situations. The core content was adapted from the HIV Connect online self-paced PrEP course for general practitioners, based on current guidelines and tailored to the identified needs of community pharmacists [10,27,29].

A few modifications were made to align the training with pharmacists' scope of practice and enhance implementation readiness. The simplified content focused on oral PrEP with tenofovir disoproxil fumarate/emtricitabine (TDF/FTC) as no other PrEP options were available or approved in Malaysia (e.g., tenofovir alafenamide/emtricitabine (TAF/FTC), long-acting injectable cabotegravir, and the dapivirine ring). A summary of special circumstances was added to aid comprehension. Additionally, brief implementation-focused modules were developed and incorporated to provide practical guidance on key aspects of the pharmacy-led PrEP service, including checklists for PrEP initiation and follow-up, adherence assessment and counselling, and referral pathways (when and who to refer) – each of which was to simplify implementation.

The final training program included 12 didactic videos (total duration: 1 hour 30 minutes), divided into PrEP-related content (63 minutes) and pharmacy-led service delivery (27 minutes) (Table 1). Videos were prepared using narrated PowerPoint slides, similar to the preparation method of HIV Connect training videos. Additional resources were provided via online access to a Google Drive folder containing local and international PrEP-related guidelines as well as information about HIV prevalence in Malaysia, HIV testing, PEP, sexually transmitted infections (STI), sexual history taking, and Sexual Orientation, Gender Identity, and Expression (SOGIE).

Table 1. Online PrEP training modules for community pharmacists and average time participants spent to complete each module.

| No. | Topics | Subtopics | Average time spent on each module (minutes: seconds) |
|---|---|---|---|
| 1. | PrEP | Introduction and indications | 5:26 |
| 2. | | PrEP options | 12:44 |
| 3. | | PrEP regimens | 7:06 |
| 4. | | Counselling | 9:08 |
| 5. | | Testing | 13:00 |
| 6. | | Same-day PrEP | 4:18 |
| 7. | | Monitoring PrEP | 3:14 |
| 8. | | Special situations | 8:56 |
| 9. | Implementation of pharmacy-led PrEP service | Project overview | 6:57 |
| 10. | | Workflow | 7:25 |
| 11. | | PrEP assessment tool | 7:30 |
| 12. | | Frequently asked questions | 6:02 |

## Participant recruitment

Participants were recruited using purposive sampling from 29 May to 4 June 2023. A total of 20 community pharmacists were initially invited to participate: 15 from the six designated implementation sites and five registered members of the MPS. Participation was voluntary, and no financial incentives were provided. Of the 20 invited, two pharmacists withdrew due to work commitments, resulting in 18 pharmacists who completed the training and evaluation components of the study.

## Training delivery

A single-arm quasi-experimental pre-post study was conducted. The training was delivered through Edpuzzle (https://edpuzzle.com), an interactive learning platform that supports progress tracking, self-paced learning, and user engagement [30]. This platform was chosen for its free access, user-friendliness and ability to ensure module completion. Participants were given two weeks to complete the training.

## Evaluation

The effectiveness of the training program was evaluated using pre- and post-training knowledge tests (S1 File), followed by a post-training feedback form (S2 File). The knowledge test items were developed based on contents of the training modules and published literature [23,25,26]. The training committee also incorporated the training needs and core competencies identified during the stakeholder consultation, such as adequate knowledge of HIV, and eligibility criteria for PrEP initiation and continuation.

The knowledge assessment consisted of 20 multiple-choice questions covering key topics, including the definition and indication of PrEP, HIV transmission, medication used for PrEP, PrEP candidacy, PrEP effectiveness and time to protection, side effects and duration of use, HIV testing and additional tests, follow-up schedules, eligibility and dosing regimen for event-driven PrEP, risk factors for kidney toxicity, counselling points, adherence, and reasons for stopping PrEP. Each correct answer was awarded five points, resulting in a maximum score of 100 points. The same set of questions was used for both pre- and post-tests to measure gain using identical items. The tests were pre-tested with five pharmacists not involved in the study to assess the clarity, comprehensibility, and appropriateness. Revisions were made based on their feedback, and the final version was reviewed and approved by the training committee.

The post-training feedback form consisted of four sections. The first section assessed key aspects of the training program, including content coverage, organization, clarity, duration, usefulness of additional resources, overall satisfaction, perceived relevance to professional practice, and likelihood of recommending the training to others. Participants rated their agreement using a five-point Likert scale, ranging from "strongly disagree" to "strongly agree". Sections two to four included open-ended questions to identify the most useful or valuable aspects of the training program, gather suggestions for improvement, and invite any additional comments or questions.

Participants accessed the training via a personalized Edpuzzle link after completing the pre-training knowledge test and demographic survey. After completing the training modules, participants completed the post-training knowledge test. They were then invited to fill out an anonymous feedback form, which was optional. All assessments and forms were administered using Google Forms, with responses automatically compiled in Google Sheets.

## Data analysis

For knowledge assessment, demographic information and pre- and post-training knowledge scores were retrieved from Google Sheets and tabulated using Microsoft Excel. Descriptive and statistical analyses were performed using SPSS

version 27.0. Paired t-tests assessed differences between pre- and post-training knowledge scores. Independent t-tests compared mean knowledge score differences by age (26–31 vs. 32–54 years), gender (male vs. female), education level (undergraduate vs. postgraduate), and prior HIV-related training (yes vs. no). One-way ANOVA examined mean knowledge score differences across ethnicity and years of experience. Statistical significance was set at $p < 0.05$.

We used an 80% correct-response threshold, which is commonly employed in assessments of clinical knowledge for pharmacists and other healthcare professionals. This benchmark is often used to ensure a high level of competency in critical areas of practice. Many pharmacy education and certification programs set an 80% passing score to certify competency. For instance, the American Society of Health-System Pharmacists (ASHP) requires a minimum score of 80% on post-assessments for its Clinical Pharmacy Basics Certificate and Research Skills Certificate programs. The Association of Clinical Research Professionals (ACRP) mandates an 80% pass rate on its Clinical Research Knowledge Assessment (CRKA) to endorse foundational competencies in clinical research ACRP, and in educational settings, an 80% threshold is often used to determine competency. For example, a study on training student pharmacists in point-of-care testing required students to achieve at least 80% on knowledge examinations to be deemed competent [31]. Last, the 80% benchmark aligns with Bloom's taxonomy, where scores above 80% indicate sufficient knowledge and positive attitudes in clinical practice assessments [32]. These examples illustrate that an 80% cutoff is a widely accepted standard to ensure that healthcare professionals possess the necessary knowledge to provide safe and effective care.

For post-training feedback, quantitative data from the Likert-scale responses were summarized using descriptive statistics. Open-ended responses from Sections 2–4 of the post-training feedback form were analyzed descriptively. Responses were grouped by common topics to identify recurring themes related to the most useful or valuable aspects of the training and suggestions for improvement. All unique responses were included, while overlapping or repetitive comments were excluded to avoid redundancy. Key insights were then summarized to support the interpretation of participant experiences and to inform future refinement of the training program.

### Ethics and consent

Ethical approval was obtained from the institutional review board at the University of Malaya Medical Centre-Medical Research Ethics Committee (UMMC-MREC) under the MREC ID: 20221026−11645. Participants were provided with a participation information sheet, and online written informed consent was obtained via Google Forms prior to study participation.

## Results

### Participant characteristics

The median age of the 18 community pharmacists was 30.5 years (IQR: 5.8). Most participants were female (77.8%), of Chinese ethnicity (89.0%), and held undergraduate degrees (89.0%). The median duration of work experience was 6.5 years (IQR: 4.5; range: 2–29 years). Only 22.2% reported prior HIV-related training or experience. Of these four participants, two had worked at a Ministry of Health (MOH) antiretroviral therapy (ART) clinic, one had completed a clinical attachment there, and one had worked with a non-governmental organization providing HIV care. On average, participants took 1 hour and 47 minutes to complete all training modules. Participant characteristics are detailed in Table 2.

### Pre- and post-training knowledge scores

Mean knowledge scores increased significantly from 64.7% pre-training to 78.9% post-training (mean difference: 14.2%; 95% CI: 8.2–20.1%; $p < 0.001$, paired t-test) (Table 3). Although most participants showed improvement, four did not, with either no change or a decline in scores. These individuals shared similar characteristics—Chinese ethnicity, an undergraduate degree, and 5–10 years of experience—suggesting that mid-career pharmacists may require more tailored or

**Table 2. Demographic and professional characteristics of participants (N = 18).**

| Participant characteristics | N (%) |
| --- | --- |
| Age (median (IQR): 30.5 (5.8) years) | |
| Range | |
| 26–31 years | 10 (55.6%) |
| 32–54 years | 8 (44.4%) |
| Gender | |
| Male | 4 (22.2%) |
| Female | 14 (77.8%) |
| Ethnicity | |
| Chinese | 16 (88.9%) |
| Malay | 1 (5.6%) |
| Indian | 1 (5.6%) |
| Highest level of education | |
| Undergraduate degree | 16 (88.9%) |
| Postgraduate master's degree | 2 (11.1%) |
| Years of experience (median (IQR): 6.5 (4.5) years) | |
| ≤5 years | 7 (38.9%) |
| 6–10 years | 9 (50.0%) |
| 11–15 years | 1 (5.6%) |
| 16–20 years | 0 (0%) |
| >20 years | 1 (5.6%) |
| Prior HIV-related experience | |
| Yes | 4 (22.2%) |
| No | 14 (77.8%) |

interactive learning strategies. Interestingly, one of these participants had prior HIV-related experience, indicating that baseline familiarity does not necessarily predict knowledge gains in self-paced online learning.

Statistical analysis revealed no significant differences in knowledge improvement by age (p = 0.302), gender (p = 0.594), ethnicity (p = 0.785), education level (p = 0.308), prior HIV-related training (p = 0.444), or years of experience (p = 0.897), suggesting the training was broadly effective across demographic groups. The observed variation in individual learning outcomes, particularly among those with similar professional profiles who did not improve, underscores the importance of offering complementary learning approaches that accommodate diverse learning styles and backgrounds.

Despite overall gains, analysis of individual test items revealed persistent knowledge gaps (Table 4). No improvement was observed in responses related to indications for PrEP or its side effects. Declines were noted in several areas, including contraindications (–6%), effectiveness (–6%), management of missed doses (–6%), identification of PrEP candidates (–12%), and counseling points (–22%). These findings suggest confusion or insufficient understanding in clinical decision-making and patient communication, which are critical areas for safe and effective PrEP delivery.

Conversely, significant gains were observed in topics initially had low baseline scores, such as time-to-protection for specific populations (+44%), eligibility for event-driven PrEP and its dosing regimen (+44% each), and HIV testing frequency (+39%). These improvements suggest the training was effective in addressing gaps in foundational knowledge. However, improvements in adherence counselling, assessment of renal toxicity risk factors, and additional tests required before initiating PrEP were modest (+5% to +11%), indicating that these topics may benefit from greater emphasis or the inclusion of more active learning strategies.

Overall, while the training improved general knowledge among participants, it did not sufficiently address all key knowledge deficits, particularly in areas related to clinical decision-making, patient assessment, and counseling. These findings

**Table 3. Participant-level pre- and post-test scores (N = 18).**

| Participant number | Pre-Test Score | Post-Test Score | Change in Score |
|---|---|---|---|
| 1 | 80 | 75 | −5 |
| 2 | 70 | 80 | +10 |
| 3 | 65 | 90 | +25 |
| 4 | 45 | 80 | +35 |
| 5 | 55 | 85 | +30 |
| 6 | 80 | 90 | +10 |
| 7 | 60 | 70 | +10 |
| 8 | 70 | 80 | +10 |
| 9 | 75 | 75 | 0 |
| 10 | 75 | 75 | 0 |
| 11 | 60 | 70 | +10 |
| 12 | 60 | 75 | +15 |
| 13 | 65 | 85 | +20 |
| 14 | 70 | 90 | +20 |
| 15 | 65 | 60 | −5 |
| 16 | 55 | 75 | +20 |
| 17 | 50 | 80 | +30 |
| 18 | 65 | 85 | +20 |
| Mean score (standard deviation, SD) | **64.7 (9.8)** | **78.8 (7.9)** | – |
| Lowest score<br>Highest score | **45**<br>**80** | **60**<br>**90** | – |
| Mean difference (95% confidence interval, CI) | **+14.2 (8.2–20.1)** | – | – |
| Percentage change | **+21.9%** | – | – |
| Paired t-test | **p < 0.001** | – | – |

suggest that community pharmacists may require more detailed explanations and enhanced instructional methods to improve comprehension in these critical domains. As such, certain content areas may benefit from more structured or interactive delivery formats, such as case-based learning and real-world clinical scenarios. Augmenting the online modules with these approaches may better equip pharmacists with the practical knowledge and skills necessary for effective PrEP service delivery.

## Participant feedback

Of the 18 participants, 13 provided anonymous feedback (Table 5). Most respondents agreed or strongly agreed that the training was relevant, well-organized, easy to follow and understand, of appropriate duration, and useful for their professional work. Overall satisfaction with the training was high, with all participants indicating they would recommend it to others. However, despite this positive perception, the findings suggest a disconnect between participants' perceived learning and their actual knowledge gains, indicating that some may have overestimated their understanding of the material.

Regarding the most useful or valuable aspects of the training program, participants appreciated the short, asynchronous format of the modules and the clarity in presenting PrEP information—including indications, types of clients to expect, and dosing regimens. They also highlighted the inclusion of a FAQ section, the summary of the PrEP service delivery process, access to supplementary materials, and the ability to revisit the content at their own pace.

**Table 4. Percentage of correct responses by question number for pre- and post-training knowledge tests.**

| Question number | Aspect of assessment | Percentage of correct responses for pre-training (%) | Percentage of correct responses for post-training (%) | Change in correct responses (%) |
|---|---|---|---|---|
| 1. | Definition of PrEP | 61 | 94 | +33 |
| 2 | Indications for PrEP | 61 | 61 | 0 |
| 3. | Medication used for PrEP | 72 | 100 | +28 |
| 4. | PrEP use in pregnancy and breastfeeding; preventive effects and duration of PrEP use | 94 | 100 | +6 |
| 5. | Contraindications for PrEP | 78 | 72 | −6 |
| 6. | Effectiveness of PrEP | 56 | 50 | −6 |
| 7. | Time-to-protection for cisgender women, transgender women on estradiol-based hormones and PWID who are on daily PrEP | 39 | 83 | +44 |
| 8. | Side effects of PrEP | 89 | 89 | 0 |
| 9. | Duration of PrEP use | 44 | 78 | +34 |
| 10. | Additional tests before starting PrEP | 78 | 89 | +11 |
| 11. | Recommended follow-up schedule | 67 | 94 | +27 |
| 12. | Frequency of HIV testing | 28 | 67 | +39 |
| 13. | Suitable candidates for event-driven PrEP | 39 | 83 | +44 |
| 14. | Dosing regimen for event-driven PrEP | 50 | 94 | +44 |
| 15. | Risk factors of renal toxicity in PrEP users | 17 | 28 | +11 |
| 16. | Counselling key points | 83 | 61 | −22 |
| 17. | PrEP adherence counselling | 89 | 94 | +5 |
| 18. | Management of missed doses for daily PrEP | 100 | 94 | −6 |
| 19. | Reasons for stopping PrEP | 94 | 100 | +6 |
| 20. | Identifying PrEP candidates | 56 | 44 | −12 |
| **Mean percentage of correct responses (SD)** | | 64.7 (23.7) | 78.8 (20.8) | – |
| **Mean difference in percentage of correct responses (95% CI)** | | | 14.0 (4.3–23.8) | |

**Table 5. Participant feedback regarding online training program (N = 13).**

| Provided statement | Strongly disagree | Disagree | Neither agree nor disagree | Agree | Strongly agree |
|---|---|---|---|---|---|
| The training covered the content I expected. | 0 | 0 | 0 | 8 (62%) | 5 (38%) |
| The content was organized and easy to follow. | 0 | 0 | 0 | 8 (62%) | 5 (38%) |
| The content was easy to understand. | 0 | 1 (8%) | 3 (23%) | 5 (38%) | 4 (31%) |
| The duration of training was appropriate. | 0 | 0 | 1 (8%) | 6 (46%) | 6 (46%) |
| The additional resources shared were helpful. | 0 | 0 | 2 (15%) | 6 (46%) | 5 (38%) |
| Overall, I am satisfied with this training. | 0 | 0 | 1 (8%) | 6 (46%) | 6 (46%) |
| This training will be useful for my work. | 0 | 0 | 0 | 7 (54%) | 6 (46%) |
| I would recommend this to others. | 0 | 0 | 0 | 6 (46%) | 7 (54%) |

Despite these positive responses, several suggestions for improvement emerged from the feedback. One participant noted that the first eight modules were fast-paced and contained a high volume of clinical information, which could be overwhelming for those new to PrEP. This raised concerns about cognitive load and the need to better scaffold complex content for learners with varying levels of familiarity.

In terms of learning preferences, several participants expressed a desire for more interactive elements. They recommended incorporating short quizzes at the end of each video to reinforce understanding and offering an indexed layout

of the training content for easier navigation and review. These suggestions reflect a broader preference for more active learning strategies to complement the self-paced format.

Participants also suggested improving clarity by adding an introductory slide to define key terms such as "cisgender," "point-of-care testing (POCT)," and "injection drug users (IDU)." A few respondents recommended including a brief overview of PEP to guide pharmacists on when to refer clients to physicians, which would enhance their confidence in making appropriate clinical decisions. Importantly, some participants expressed uncertainty about managing PrEP for clients on estradiol or hormone therapy, as well as how to navigate conversations with transgender clients.

Notably, many of these feedback points reflect areas in which knowledge scores were lower or declined, such as counseling, identifying PrEP candidates, and understanding nuanced clinical content. This convergence between self-reported challenges and test performance highlights a need to enhance these components in future training iterations through more targeted content, inclusive language, and interactive learning strategies.

## Discussion

To prepare community pharmacists to competently provide a new service (i.e., PrEP), they must first be equipped with essential clinical knowledge. To our knowledge, this is the first study to adapt a physician-targeted PrEP training program for pharmacists and evaluate its effectiveness, demonstrating a significant improvement in knowledge scores (from 64.7% to 78.9%, p < 0.001).

Several studies have assessed pharmacists' PrEP knowledge, though none have included training followed by pre-and post-assessment. A study in Utah reported a mean PrEP knowledge score of 4.99 (SD 2.1) among 251 pharmacists, with scores ranging from 1 to 8 for correct responses [23]. Another study in Zimbabwe found that 58% of pharmacists (65/112) were knowledgeable about PrEP, scoring 9 or higher out of 15 questions [33]. Meanwhile, a study of among 194 U.S. pharmacy students reported a mean knowledge score of 78.8% [26]. Although these studies used different surveys and did not establish a standardized cut-off for PrEP competency, we used a benchmark of 80% for pharmacists who undergo structured PrEP training, common in e-learning modules.

This threshold is intended to reflect a level of knowledge above baseline levels observed among pharmacy students, and to approximate a level of readiness for providing PrEP-related services. However, we acknowledge that this threshold is not evidence-based and further research is needed to define and validate an appropriate competency standard for pharmacists involved in PrEP service delivery.

In this study, statistical analyses showed no significant differences in knowledge improvement by age, gender, ethnicity, education level, years of experience or prior HIV-related training or experience. This finding aligns with the study conducted in Zimbabwe, which reported no significant associations between PrEP knowledge and demographic factors, such as age, gender, years of practice, and pharmacy setting [33]. In contrast, the U.S. study found that cisgender MSM, bisexual participants, and White students had higher PrEP knowledge than their peers, and that late-phase students outperformed early-phase students [26]. Similarly, a study in Utah reported higher PrEP knowledge among pharmacists with PharmD degrees and fewer years of experience [23]. These differences may reflect contextual variations in education, training, or baseline familiarity with HIV prevention services, including PrEP, highlighting the potential of structured training programs to reduce disparities in knowledge across diverse pharmacist populations.

While the online training program was generally effective for most participants, as evidenced by overall improvement in knowledge scores, it did not produce the desired consistent knowledge gains across all participants. Four participants did not demonstrate any improvement, and no gains were observed in topics such as PrEP indications and side effects.

Certain knowledge gaps persisted despite training. While all participants identified the approved PrEP medication post-training, knowledge of HIV testing frequency improved from 28% to 67%; however, correct responses on contraindications declined from 78% to 72%. A study in New York found that only 38% of pharmacy students knew the recommended testing frequency and 28% recognized reduced creatinine clearance as a contraindication to

prescribing tenofovir-based PrEP [25]. These findings suggest that applied clinical reasoning remains a challenge, even when foundational knowledge is covered.

Knowledge deficits also remained in key areas, such as effectiveness, counselling key points, management of missed doses, and identification of PrEP candidates. This indicates that while the training effectively conveyed foundational concepts, it was less successful in fostering applied clinical reasoning, which is an essential skill for assessing PrEP suitability and counseling diverse clients.

Participant feedback provided valuable insight into knowledge gaps. Several participants reported difficulties with the pacing and density of clinical content in the initial modules, which may have contributed to lower post-training performance in topics such as counseling, and identification of eligible clients. Suggestions to include introductory slides for key terms, overviews of related topics such as PEP, and clearer guidance on managing transgender clients aligned closely with the test items that showed decreased or modest gains. These reflections indicate that participants not only struggled with specific clinical knowledge but also required more support in applying that knowledge to real-world scenarios.

Participants also called for more interactive features, such as short quizzes after subtopics and indexed content. This aligns with evidence supporting test-enhanced learning, which improves retention and understanding [34,35]. Moreover, participants expressed uncertainty around managing PrEP for transgender clients and integrating PEP into counseling—areas directly reflected in lower-scoring test items. The convergence of knowledge gaps and qualitative feedback emphasizes the need for improved instruction on inclusive care, clinical decision-making, and patient communication.

The training program was intentionally designed as a self-paced, asynchronous online course to accommodate the time constraints and staffing challenges commonly encountered in community pharmacy settings. This flexible delivery format was well-received by participants, who appreciated the ability to engage with the material at their convenience without disrupting daily workflow. The self-directed nature of the program, however, may have inadvertently contributed to a gap between participants' perceived learning and their actual knowledge acquisition.

This discrepancy aligns with findings from a systematic review on pharmacists' engagement with e-learning platforms, which revealed a recurring pattern: although most users expressed high levels of satisfaction—highlighting advantages such as flexibility, user-friendliness, and relevance to practice—these subjective assessments did not consistently correlate with measurable improvements in knowledge, skill acquisition, or clinical performance [36]. While the review affirmed that e-learning can be just as effective as traditional face-to-face instruction for knowledge delivery, it also underscored a key limitation: when used in isolation, digital platforms often fall short in cultivating applied skills such as patient counseling, therapeutic decision-making, or inter-professional communication. These findings reinforce the importance of integrating e-learning with interactive elements—such as live discussions, case-based simulations, or hands-on workshops—to better translate theoretical knowledge into real-world clinical competence [36].

According to the Expert Recommendations for Implementing Change (ERIC) taxonomy and implementation science literature, while training and education are essential strategies, they are often insufficient on their own and should be supplemented with strategies, such as audit and feedback, educational outreach, and external facilitation to strengthen implementation [37,38]. Future efforts could incorporate follow-up strategies such as feedback on PrEP case assessments, interactive or tailored content, and regular check-ins or mentoring by program implementers, to help reinforce learning, particularly for mid-career pharmacists who may derive less benefit from conventional e-learning formats.

A systematic review in medical education found that modern learning techniques, such as case-based, problem-based, and simulation-based learning, were more effective than traditional didactic lectures in improving knowledge acquisition and learner engagement [39]. These approaches promote active learning, enhance clinical reasoning, support knowledge retention, and develop essential analytical and problem-solving skills [39].

A one-day in-person workshop was conducted two weeks after completion of the online modules as a reinforcement activity to clarify complex topics and strengthen pharmacists' readiness for PrEP service delivery. All 18 pharmacists who completed the online training were invited to attend and 16 attended. The workshop was facilitated by

a multidisciplinary team, including infectious disease physicians, academicians, and community health workers from NGOs, and was attended by a total of 35 participants, including pharmacists. It aimed to build practical skills through applied learning methods such as case-based discussions, interactive group activities, and role-playing exercises. Key topics included effective communication, stigma reduction, and appropriate referral pathways to clinical and community-based services. Role-playing sessions, facilitated by NGO representatives, were specifically designed to enhance pharmacists' sensitivity to the needs of key populations. To further reinforce learning, we incorporated a review of knowledge assessment items, particularly those with lower post-training correct responses. These activities helped participants revisit key concepts, clarify misconceptions, and apply their knowledge to realistic scenarios.

To support real-world readiness and continued learning, we conducted a dry-run session prior to implementation, simulating client interactions to give pharmacists the opportunity to practice delivering the PrEP service as intended. Additionally, a WhatsApp group was created to enable real-time communication, feedback, and peer support throughout the implementation phase. Coordinated by the research team, this platform supported informal mentoring, addressed emerging questions, and facilitated continuous learning. These active learning strategies proved valuable in bridging theoretical knowledge with practice and demonstrated the potential of blended training models to support the scale-up of pharmacy-led PrEP services.

Current evidence shows that pharmacy-led PrEP services have been implemented or piloted in several countries, widely accepted by clients and pharmacy providers and shown to reach underserved populations, such as young men and minorities, thereby improving overall PrEP coverage [21,40]. With proper training and oversight, pharmacists can safely initiate and manage clients on PrEP, achieving initiation and continuation rates comparable to those of traditional clinics [40]. This presents an opportunity for community pharmacists in Malaysia to expand their role in public health and contribute to the national goal of achieving zero new infections outlined in the Malaysian NSP to End AIDS by 2030 [5]. The online training program could be a viable tool to support this initiative; however, improvements in instructional design, delivery format, and system-level integration are essential for sustained impact.

Scaling up the training program for wider implementation will also require addressing practical barriers such as internet access, digital literacy, and varying levels of baseline knowledge. Pharmacists in rural or underserved areas may face connectivity challenges or have limited prior exposure to HIV-related services, potentially affecting their learning outcomes. Downloadable modules, mobile-optimized platforms, and regionally organized in-person workshops could help bridge these gaps. Integrating pharmacists into the national PrEP service framework will require policy and regulatory support, including a formal certification pathway, endorsement of standardized training, and mechanisms for collaboration between pharmacists and HIV physicians via telemedicine platforms.

## Strengths

This is the first study in Malaysia, and among the first in the Asia-Pacific, to develop and evaluate a pharmacist-specific online PrEP training program. The training was informed by the EPIS implementation science framework, ensuring that contextual, structural, and practical considerations were integrated into its design. Unlike previous cross-sectional studies, we assessed knowledge improvement pre- and post-training. Additionally, incorporating participant feedback provided valuable insight into the learning experience and highlighted areas for improvement. Stakeholder consultation during training development ensured that the content was contextually relevant, feasible, and aligned with national HIV prevention priorities, enhancing the potential for scale-up and policy integration.

## Limitations

Despite the many important findings, there are limitations. First, the small, relatively homogenous sample may not accurately represent the broader population of Malaysian community pharmacists. Most participants were of Chinese ethnicity, held undergraduate degrees, and had about 8 years of experience. While the sample was representative of private community pharmacists in Malaysia [41], future studies should aim for broader representation. Second, participation was

limited to pharmacists from study sites and MPS representatives in Klang Valley due to budget constraints. Findings here may not reflect community pharmacists in other regions of the country.

Third, knowledge was assessed immediately after the online training, but not after the in-person workshop. Since the workshop incorporated active learning strategies, an additional post-workshop assessment could have helped evaluate the added value of these components. While the workshop was not formally evaluated, the workshop served as a complementary component to support applied learning and readiness for pharmacy-led PrEP service delivery. Fourth, the knowledge test items used in this study were not validated through psychometric testing. Although they were developed based on training content and supported by published literature, the absence of formal validation may limit their reliability and comparability to other instruments.

Fifth, performance bias was possible. Participants were aware of the pre-post design and may have reviewed materials in advance. Also, the online testing format allowed for potential access to external resources despite instructions to complete the tests independently, which could inflate scores. Additionally, although the feedback forms were completed anonymously, social desirability bias may still have influenced participants' responses, potentially leading to an overestimation of satisfaction or perceived usefulness of the training.

Sixth, we measured only short-term knowledge gains and long-term retention was not assessed. Other studies have shown improved knowledge retention after 30 days [42], suggesting that practical application over time may enhance retention and that follow-up assessments are important for evaluating long-term impact. Lastly, we did not assess participants' confidence or counseling skills. While knowledge acquisition is a necessary foundation, it does not guarantee competence in delivering PrEP services. Future studies should evaluate communication skills, self-efficacy, and observed performance to better capture readiness for real-world service delivery.

## Conclusions

This novel online, self-paced PrEP training program, developed to prepare community pharmacists for the future implementation of pharmacy-led PrEP service delivery, effectively improved PrEP knowledge. However, variability in knowledge gains and persistent knowledge gaps, particularly in areas such as patient counseling, eligibility assessment, contraindications, effectiveness and management of missed doses, highlight the need for further refinement. Incorporating active learning strategies such as case-based, problem-based, and simulation-based learning within a blended training model may enhance comprehension and support the translation of theoretical knowledge into practical application, thereby strengthening clinical decision-making and counseling skills essential for confident and effective PrEP service delivery.

Future studies should involve larger and more diverse samples to enhance generalizability and statistical power, utilize validated knowledge assessment tools, and examine predictors of PrEP knowledge improvement. In addition, extended follow-up is warranted to evaluate sustained knowledge retention and its translation into real-world practice. Evaluating the added value of blended training components—such as in-person workshops, peer support mechanisms, and structured feedback—will also be important to understand their impact on pharmacist confidence, counseling performance, and readiness to deliver PrEP services effectively. Ultimately, this training represents a critical step toward integrating pharmacists into national PrEP service delivery models and advancing equitable access to HIV prevention in Malaysia.

## Supporting information

**S1 File. Pre- and post-training knowledge test questionnaire.**
(DOCX)

**S2 File. Post-training feedback form.**
(DOCX)

## Acknowledgments

We would like to thank all research participants for their commitment and willingness to participate in this study. We are grateful for the invaluable support from the Centre of Excellence for Research in Infectious Diseases and AIDS (CERiA), the Malaysian Society for HIV Medicine (MASHM), and the Malaysian Pharmacists Society (MPS) throughout the conduct of this study. We also wish to acknowledge the contribution of the research assistants involved in the preparation and execution of the online training, namely: Foo Xu Qing and Assif Shamim Mustaffa.

## Author contributions

**Conceptualization:** Yan Nee Gan, Rafdzah Zaki, Frederick L. Altice, Sin How Lim, Iskandar Azwa.

**Data curation:** Yan Nee Gan.

**Formal analysis:** Yan Nee Gan, Safia Alia Hafidzar.

**Funding acquisition:** Yan Nee Gan, Frederick L. Altice, Iskandar Azwa.

**Investigation:** Yan Nee Gan.

**Methodology:** Yan Nee Gan, Rafdzah Zaki, Kwee Choy Koh, Mariani Ahmad Nizaruddin, Rosnida Mohd Noh, Khairil Erwan Khalid, Andrew Yap, Frederick L. Altice, Sin How Lim, Iskandar Azwa.

**Project administration:** Yan Nee Gan, Rafdzah Zaki, Frederick L. Altice, Sin How Lim, Iskandar Azwa.

**Resources:** Kwee Choy Koh, Mariani Ahmad Nizaruddin, Rosnida Mohd Noh, Khairil Erwan Khalid, Andrew Yap, Iskandar Azwa.

**Supervision:** Rafdzah Zaki, Frederick L. Altice, Sin How Lim, Iskandar Azwa.

**Validation:** Yan Nee Gan, Kwee Choy Koh, Mariani Ahmad Nizaruddin, Rosnida Mohd Noh, Khairil Erwan Khalid, Andrew Yap, Frederick L. Altice, Iskandar Azwa.

**Visualization:** Yan Nee Gan, Safia Alia Hafidzar.

**Writing – original draft:** Yan Nee Gan, Safia Alia Hafidzar.

**Writing – review & editing:** Yan Nee Gan, Rafdzah Zaki, Safia Alia Hafidzar, Kwee Choy Koh, Mariani Ahmad Nizaruddin, Rosnida Mohd Noh, Khairil Erwan Khalid, Andrew Yap, Frederick L. Altice, Sin How Lim, Iskandar Azwa.

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
