## [Decision Letter · Decision Letter 0]

13 Feb 2025

PONE-D-24-57343Development and delivery of an online HIV pre-exposure prophylaxis (PrEP) training for community pharmacists to prepare for the implementation of a pharmacy-led PrEP service in Malaysia.PLOS ONE

Dear Dr. Zaki,

Thank you for submitting your manuscript to PLOS ONE. After careful consideration, we feel that it has merit but does not fully meet PLOS ONE’s publication criteria as it currently stands. Therefore, we invite you to submit a revised version of the manuscript that addresses the points raised during the review process.

We look forward to receiving your revised manuscript.

Kind regards,

Ali Ahmed, PhD

Academic Editor

PLOS ONE

Journal requirements:   When submitting your revision, we need you to address these additional requirements. 1. Please ensure that your manuscript meets PLOS ONE's style requirements, including those for file naming. The PLOS ONE style templates can be found at https://journals.plos.org/plosone/s/file?id=wjVg/PLOSOne_formatting_sample_main_body.pdf and https://journals.plos.org/plosone/s/file?id=ba62/PLOSOne_formatting_sample_title_authors_affiliations.pdf. 2. Please upload a copy of Supplemental Information File 1 and Supplemental Information File 2 to which you refer in your text on page 9. Please amend the file type to 'Supporting Information'. If the Supplementary file is no longer to be included as part of the submission please remove all reference to it within the text.

Additional Editor Comments:

I have only concern of sample diversity as most of participants are chinese, given the diversity of Malaysia. Does this impact the implications of your findings?

Reviewers' comments:

Reviewer's Responses to Questions

**Comments to the Author**

1. Is the manuscript technically sound, and do the data support the conclusions?

Reviewer #1: Partly

Reviewer #2: Yes

Reviewer #3: Partly

2. Has the statistical analysis been performed appropriately and rigorously? 

Reviewer #1: I Don't Know

Reviewer #2: Yes

Reviewer #3: Yes

3. Have the authors made all data underlying the findings in their manuscript fully available?

Reviewer #1: Yes

Reviewer #2: Yes

Reviewer #3: Yes

4. Is the manuscript presented in an intelligible fashion and written in standard English?

Reviewer #1: No

Reviewer #2: Yes

Reviewer #3: Yes

5. Review Comments to the Author

Reviewer #1: Consider revising the title - Development and delivery of an online HIV pre-exposure prophylaxis (PrEP) training for community pharmacists to prepare for the implementation of a pharmacy-led PrEP service in Malaysia. This title could be made more explicit e.g., Development and Evaluation of an Online PrEP Training Program for Community Pharmacists to Enhance Pharmacy-Led HIV Prevention Services in Malaysia

Abstract- The phrases "stigmatized key populations" and "non-traditional differentiated service delivery models, like pharmacies" are slightly repetitive when describing the rationale. Simplify for conciseness. While informative, the committee, platform, and timeline description could be condensed. Focus on the core aspects, such as the training content, evaluation tools, and key methodological steps. While the results highlight the improvement in knowledge scores, the discussion of gaps in knowledge and variability among participants is not explored sufficiently. Briefly mention specific gaps to better contextualize the need for enhanced training methods. Link participant demographics to their potential impact on training outcomes. While the conclusion mentions the need for modern methodologies, it does not connect how these would address the identified gaps.

Introduction- While the introduction thoroughly describes the HIV epidemic in Malaysia, key affected populations, and existing service delivery gaps, it is overly detailed, with some points repeated unnecessarily, e.g., barriers to public healthcare facilities and the potential of pharmacy-based models. The transition between some paragraphs feels abrupt, particularly when discussing global recommendations, local implementation, and the rationale for pharmacy-led models. The introduction outlines the problem well but doesn't sufficiently emphasize how the study addresses these gaps.

The Materials and Methods section provides detailed descriptions of the study's design, processes, and tools, which is commendable. The EPIS framework adds rigor and aligns the study within a recognized implementation science framework, strengthening the study's credibility. However, the section is densely packed with information, making it challenging to follow. Key components like "Development of the training program," "Delivery," and "Data analysis" could be more distinct. Breaking the section into clearly labeled subsections, e.g., Training Program Development, Study Design, Participant Recruitment, Training Delivery, Evaluation, and Data Analysis, would improve readability. Certain details, e.g., repeated mentions of module content and pre/post-test descriptions, are unnecessarily repeated. While the stakeholder consultation is mentioned, its contributions to the training program development are not elaborated upon. The author should highlight specific insights or recommendations from stakeholders that informed the program design to strengthen the relevance of the consultation. There is insufficient explanation of why particular tools, e.g., Edpuzzle and Google Forms, were chosen. The section uses long paragraphs that make key details hard to locate.

Results -The results section contains substantial data organized around demographic characteristics, pre- and post-training knowledge scores, feedback on the training program, and participant suggestions for improvement. Demographic information is detailed but repetitive, with percentages mentioned in both text and table form. Summarize key demographic trends in the text and refer readers to the table for further details. Table 3 presents data participant-by-participant, which may not be necessary for interpretation. Condense Table 3 to highlight summary statistics (mean, SD, minimum, maximum, p-value) without individual scores. The narrative describing pre- and post-training knowledge scores is overly general and does not critically engage with the data. For example, the statement “the online training appears insufficient” is unsupported by nuanced analysis. Provide a more detailed interpretation of the data. The feedback is summarized well but lacks a deeper synthesis or connection to broader findings. Authors need to analyze feedback to identify recurring themes, e.g., clarity of clinical content, need for quizzes, the pacing of modules, etc, and link these themes to observed knowledge gaps.

Discussion section—While improving knowledge scores is significant, the discussion could delve deeper into how these scores translate into practical application or readiness to provide PrEP services. The discussion briefly mentions areas where the training fell short, e.g., indications, side effects, and contraindications, but it does not explore why these gaps persist or propose specific ways to address them in future iterations. While participant feedback is mentioned, the discussion does not thoroughly analyze or contextualize it. For example, the perceived disconnect between participants’ satisfaction and actual knowledge gains could be explored in more depth. The discussion does not sufficiently address potential barriers to implementing the training program on a larger scale, such as resource constraints, internet accessibility, or variations in pharmacists' baseline knowledge. While the study's relevance to public health goals is noted, e.g., achieving zero new HIV infections in Malaysia by 2030, the practical steps needed to integrate pharmacists into the national PrEP service framework are not discussed. Although the ERIC taxonomy is cited, its application in this context is not detailed. Explaining how ERIC strategies could specifically enhance pharmacist training would strengthen the argument.

Conclusions—The conclusions summarize the study's key findings, emphasizing the novelty and effectiveness of the online PrEP training program in increasing pharmacists' knowledge. However, there is an overgeneralization. For example, while the training program improved knowledge, the conclusions imply broader success without fully addressing the uneven knowledge gains among participants or the remaining critical knowledge gaps. Recommendations for incorporating advanced learning methodologies are mentioned but lack specificity. For example, how case-based or simulation-based learning could be practically integrated is not explored. The conclusions do not discuss potential logistical or systemic challenges, e.g., funding, time constraints, or accessibility issues that might hinder nationwide program implementation. There is little mention of how increased pharmacist knowledge translates into actual improvements in PrEP service delivery or uptake in Malaysia. While knowledge improvement is highlighted, confidence and skill in counseling and service provision are not sufficiently addressed, even though they are critical for successful PrEP delivery.

Reviewer #2: Congratulations to the authors for the well written manuscript. The manuscript highlights a salient issue on the development and delivery of online PrEP training for new cadres such as pharmacy providers in the context of task shifting services. Two issues for consideration:

1. The title of the manuscript gives prominence to the development and delivery of the online training. However, the primary substance of the manuscript is on PrEP knowledge gains from undertaking the training. Would the authors consider reframing their title to address the primary focus of their manuscript.

2. The limitations of the study are well described but no strengths are included. It would be good to read why the authors think this research adds to the field and why their study is a strong one.

Reviewer #3: OVERALL: This manuscript focuses on development and assessment of an online, self-paced PrEP training program for community pharmacist for PrEP delivery in Malaysia. This is the first known assessment of such a training program in Malaysia and adds to the sparse literature on how pharmacy providers could be trained to deliver PrEP services in Malaysia and other global settings. While addressing an important topic, some further details on how the training was developed and what went into development of the assessment tools would be helpful. Additionally, it isn’t clear why the authors choose to only describe the online component of this training program and not the in-person training that followed—which seemed to utilize many of the training strategies the authors suggest for enhancement of the online training module. I think presenting both training components together would enhance this paper and present a wholistic package of how pharmacy providers might be trained on PrEP delivery. Further, both the Introduction and Discussion of the paper could be revised for clarity and focus.

MAJOR:

• (Abstract, Results): It isn’t clear how to interpret the “knowledge score”. How was this constructed and scored; what types of questions were included in this assessment?

• (Abstract, Results): It is not totally clear what you mean from the following sentence: “While training significantly improved knowledge scores, it did not produce consistent knowledge gains for all participants or address some critical PrEP knowledge gaps”. Can you be more specific on the gaps of interest? And how many participants didn’t have knowledge gains?

• (Abstract, Conclusions): Might temper conclusions because not proving in this abstract that pharmacy PrEP delivery increases PrEP access.

• (Introduction): The introduction could be abbreviated for focus and clarity.

• (Methods): How was the content of the open-ended questions analyzed? Which subsequently led to key themes on the most useful aspects of the training and areas for improvement presented in the Results?

• (Discussion, paragraph 2): Still not clear where your suggested threshold of 80% knowledge comes from. Seems arbitrary to suggest such threshold if different knowledge assessments using different tools.

• (Methods): Details on how pharmacists were trained on HIV testing required for PrEP delivery missing. Especially since this might require in-person practice that is difficult to delivery remotely.

• (Discussion): Unclear what you would measure the effect of training on? If using as an ERIC implementation strategy, wouldn’t the ultimate outcome be PrEP uptake at the pharmacy versus knowledge among providers? And you did see a positive change in knowledge among providers post-training, so unclear why you are suggesting that the training wasn’t effective at knowledge gain?

• (Discussion): You note that modern learning methodologies (e.g., case-based, team-based) could improve knowledge outcomes among pharmacy providers – in the methods, might be worth mentioning why you didn’t incorporate any of these approaches when you developed this online training.

• (Discussion): IF the online training was just part one of provider training, followed by the in-person training – why is the in-person training not included in this manuscript to complete the discussion of how you might train pharmacy providers for PrEP delivery?

• (Discussion & limitations): These sections of the paper are long and could be abbreviated for focus and clarity.

MINOR:

• (Formatting): Formatting of the current manuscript is a little confusing – the distinction between sections and subsections is not clear. Additional subsections (e.g., data collection, analysis) in the methods could enhance clarity.

• (Methods): The April 2023 stakeholder meeting is mentioned without details on what types of stakeholders attended this meeting and what was discussed. Adding a brief description of this would be helpful.

• (Methods, Delivery of the online self-paced PrEP training program): Would suggest moving several details here (e.g., who attended the training, duration of engagement) to the Results section.

• (Results, Table 3): Not sure inclusion of this table in the main paper is necessary – would suggest including as an appendix, if want to keep.

• (Results, Table 4): Would consider adding a column that is the % change from pre- to pos-training assessment. Could include overall score across question at bottom of this table (i.e., info in Table 3).

6. PLOS authors have the option to publish the peer review history of their article (what does this mean? ). If published, this will include your full peer review and any attached files.

**Do you want your identity to be public for this peer review?** For information about this choice, including consent withdrawal, please see our Privacy Policy .

Reviewer #1: No

Reviewer #2: **Yes: ** Daniel Were

Reviewer #3: No

---

## [Author Response · Author response to Decision Letter 1]

30 Apr 2025

Dr. Rafdzah Ahmad Zaki

Public Health Medicine Specialist and Professor,

Department of Social and Preventive Medicine,

Faculty of Medicine

Universiti Malaya, Malaysia

rafdzah@ummc.edu.my

25th April 2025

Editor-in-Chief

PLOS ONE

Subject: Rebuttal Letter for Manuscript Revision (PONE-D-24-57343)

Dear Dr. Ali Ahmed,

We sincerely appreciate the opportunity to revise our manuscript, titled "Development and delivery of an online HIV pre-exposure prophylaxis (PrEP) training for community pharmacists to prepare for the implementation of a pharmacy-led PrEP service in Malaysia" (Manuscript ID: PONE-D-24-57343).

We are grateful for the constructive feedback provided by the reviewers and the academic editor. We have carefully addressed each of the comments and have made the necessary revisions to enhance the clarity and quality of our manuscript. Below, we provide a point-by-point response to each comment, detailing the changes made.

Comment: Please ensure that your manuscript meets PLOS ONE's style requirements, including those for file naming. The PLOS ONE style templates can be found at

Response: We have reviewed and updated our manuscript to ensure compliance with PLOS ONE’s style requirements, including formatting and file naming conventions. The revised manuscript follows the guidelines provided in the PLOS ONE style templates.

Comment: Please upload a copy of Supplemental Information File 1 and Supplemental Information File 2 to which you refer in your text on page 9. Please amend the file type to 'Supporting Information'. If the Supplementary file is no longer to be included as part of the submission, please remove all reference to it within the text.

Response: We have uploaded Supplemental Information File 1 and Supplemental Information File 2 as 'Supporting Information' files, as per the journal’s requirements. It was initially included in the manuscript under S1 Appendix and S2 Appendix. Additionally, we have ensured that all references to these files within the manuscript are accurate.

Response to the Academic Editor’s Comments:

Comment: I have only concern of sample diversity as most of participants are Chinese, given the diversity of Malaysia. Does this impact the implications of your findings?

Response: We acknowledge this concern and have addressed it in the manuscript. The predominance of Chinese pharmacists in our sample reflects the demographic distribution in private pharmacies in Malaysia. However, the training program was adapted from local PrEP training for general practitioners and follows local guidelines and core PrEP competencies applicable to all pharmacists. Future efforts will focus on broader outreach to enhance diversity in participation.

Response to Reviewer 1’s Comments:

Comment 1: Consider revising the title - Development and delivery of an online HIV pre-exposure prophylaxis (PrEP) training for community pharmacists to prepare for the implementation of a pharmacy-led PrEP service in Malaysia. This title could be made more explicit e.g., Development and Evaluation of an Online PrEP Training Program for Community Pharmacists to Enhance Pharmacy-Led HIV Prevention Services in Malaysia.

Response: We agree and revised the title to “Development and evaluation of an online HIV pre-exposure prophylaxis (PrEP) training program for community pharmacists to implement pharmacy-led PrEP services in Malaysia”.

Comment 2: Abstract - The phrases "stigmatized key populations" and "non-traditional differentiated service delivery models, like pharmacies" are slightly repetitive when describing the rationale. Simplify for conciseness. While informative, the committee, platform, and timeline description could be condensed. Focus on the core aspects, such as the training content, evaluation tools, and key methodological steps. While the results highlight the improvement in knowledge scores, the discussion of gaps in knowledge and variability among participants is not explored sufficiently. Briefly mention specific gaps to better contextualize the need for enhanced training methods. Link participant demographics to their potential impact on training outcomes. While the conclusion mentions the need for modern methodologies, it does not connect how these would address the identified gaps.

Response: The abstract has been refined to enhance clarity and remove redundant phrases while maintaining key information on training content, evaluation, and methodological aspects.

Comment 3: Introduction - While the introduction thoroughly describes the HIV epidemic in Malaysia, key affected populations, and existing service delivery gaps, it is overly detailed, with some points repeated unnecessarily, e.g., barriers to public healthcare facilities and the potential of pharmacy-based models. The transition between some paragraphs feels abrupt, particularly when discussing global recommendations, local implementation, and the rationale for pharmacy-led models. The introduction outlines the problem well but doesn't sufficiently emphasize how the study addresses these gaps.

Response: We have streamlined the introduction to reduce redundancy and tightened the narrative, particularly around barriers to existing service delivery and the rationale for pharmacy-based models. Transitions between paragraphs have been improved to ensure a more coherent flow from global recommendations to local implementation efforts. We have also revised the final paragraph to more explicitly state how our study addresses the identified gaps by developing and piloting an online PrEP training module specifically for community pharmacists—an essential step toward enabling future pharmacy-led PrEP service delivery in Malaysia.

Comment 4: The Materials and Methods section provides detailed descriptions of the study's design, processes, and tools, which is commendable. The EPIS framework adds rigor and aligns the study within a recognized implementation science framework, strengthening the study's credibility. However, the section is densely packed with information, making it challenging to follow. Key components like "Development of the training program," "Delivery," and "Data analysis" could be more distinct. Breaking the section into clearly labeled subsections, e.g., Training Program Development, Study Design, Participant Recruitment, Training Delivery, Evaluation, and Data Analysis, would improve readability. Certain details, e.g., repeated mentions of module content and pre/post-test descriptions, are unnecessarily repeated. While the stakeholder consultation is mentioned, its contributions to the training program development are not elaborated upon. The author should highlight specific insights or recommendations from stakeholders that informed the program design to strengthen the relevance of the consultation. There is insufficient explanation of why particular tools, e.g., Edpuzzle and Google Forms, were chosen. The section uses long paragraphs that make key details hard to locate.

Response: We agree and have included subsections to improve clarity and reduced redundancy. The use of Edpuzzle and Google Forms are not necessarily salient here, but we include them to ensure that readers understand which methods we selected – mostly for ease when interacting with participants. We do, however, provide more explanation for how these platforms work and why we used them.

Comment 5: Results -The results section contains substantial data organized around demographic characteristics, pre- and post-training knowledge scores, feedback on the training program, and participant suggestions for improvement. Demographic information is detailed but repetitive, with percentages mentioned in both text and table form. Summarize key demographic trends in the text and refer readers to the table for further details. Table 3 presents data participant-by-participant, which may not be necessary for interpretation. Condense Table 3 to highlight summary statistics (mean, SD, minimum, maximum, p-value) without individual scores. The narrative describing pre- and post-training knowledge scores is overly general and does not critically engage with the data. For example, the statement “the online training appears insufficient” is unsupported by nuanced analysis. Provide a more detailed interpretation of the data. The feedback is summarized well but lacks a deeper synthesis or connection to broader findings. Authors need to analyze feedback to identify recurring themes, e.g., clarity of clinical content, need for quizzes, the pacing of modules, etc, and link these themes to observed knowledge gaps.

Response: We appreciate the reviewer’s detailed feedback and have revised the Results section accordingly. To reduce redundancy, we have summarized key demographic trends in the text while referring readers to the table for details. In Table 3, we have added a column showing the percentage change in scores and the overall percentage change across participants. We believe retaining this table in the main manuscript is important to illustrate individual pre- and post-training scores, especially since four participants did not show improvement after training. Additionally, we have refined the discussion of knowledge scores to provide a more critical interpretation of the data, ensuring conclusions are supported by nuanced analysis.

To provide a more critical engagement with the data and to address the reviewer’s suggestion to link participant demographics to training outcomes, we conducted additional statistical analyses that were not part of the original manuscript. Specifically, independent t-tests were used to examine differences in post-training knowledge scores by gender (male vs. female), education level (undergraduate vs. postgraduate), and prior HIV-related training (yes vs. no), while one-way ANOVA was used to compare scores across age groups, ethnicities, and years of professional experience. These analyses are now described in the Data Analysis subsection of the Methods, with results reported briefly in the Results section. No statistically significant differences were found across these demographic variables, suggesting that the online training program was broadly effective regardless of participants’ background characteristics. However, this finding should be interpreted with caution due to the small sample size, which we have acknowledged as a limitation in the revised manuscript.

The statement on training sufficiency has been revised for accuracy. We have also expanded the synthesis of participant feedback, identifying recurring themes such as clarity of clinical content, the need for interactive elements (e.g., quizzes), and module pacing, linking these to observed knowledge gaps for a more comprehensive interpretation.

Comment 6: Discussion section—While improving knowledge scores is significant, the discussion could delve deeper into how these scores translate into practical application or readiness to provide PrEP services. The discussion briefly mentions areas where the training fell short, e.g., indications, side effects, and contraindications, but it does not explore why these gaps persist or propose specific ways to address them in future iterations. While participant feedback is mentioned, the discussion does not thoroughly analyze or contextualize it. For example, the perceived disconnect between participants’ satisfaction and actual knowledge gains could be explored in more depth. The discussion does not sufficiently address potential barriers to implementing the training program on a larger scale, such as resource constraints, internet accessibility, or variations in pharmacists' baseline knowledge. While the study's relevance to public health goals is noted, e.g., achieving zero new HIV infections in Malaysia by 2030, the practical steps needed to integrate pharmacists into the national PrEP service framework are not discussed. Although the ERIC taxonomy is cited, its application in this context is not detailed. Explaining how ERIC strategies could specifically enhance pharmacist training would strengthen the argument.

Response: We agree and have expanded the Discussion section to better contextualize how improved knowledge scores translate into pharmacists' readiness to provide PrEP services. We have explored persistent knowledge gaps, particularly in indications, side effects, and contraindications, and proposed strategies such as case-based learning. Additionally, we have noted that in-person training included a review and discussion of questions with lower correct response rates. We have also analysed the disconnect between participant satisfaction and actual knowledge gains and expanded discussions on barriers to scaling up the training, including resource constraints and baseline knowledge variations. To strengthen relevance to national public health goals, we have outlined practical steps for integrating pharmacists into Malaysia’s PrEP service framework, including policy support, pharmacist certification pathways, and collaboration with healthcare providers. Finally, we have elaborated on the application of ERIC strategies that might be useful to enhance implementation.

Comment 7: Conclusions—The conclusions summarize the study's key findings, emphasizing the novelty and effectiveness of the online PrEP training program in increasing pharmacists' knowledge. However, there is an overgeneralization. For example, while the training program improved knowledge, the conclusions imply broader success without fully addressing the uneven knowledge gains among participants or the remaining critical knowledge gaps. Recommendations for incorporating advanced learning methodologies are mentioned but lack specificity. For example, how case-based or simulation-based learning could be practically integrated is not explored. The conclusions do not discuss potential logistical or systemic challenges, e.g., funding, time constraints, or accessibility issues that might hinder nationwide program implementation. There is little mention of how increased pharmacist knowledge translates into actual improvements in PrEP service delivery or uptake in Malaysia. While knowledge improvement is highlighted, confidence and skill in counseling and service provision are not sufficiently addressed, even though they are critical for successful PrEP delivery.

Response: We agree and have revised the Conclusions section to avoid over-generalization and ensure a more balanced interpretation of the findings. We now acknowledge the variability in knowledge gains among participants and highlight remaining critical knowledge gaps. Additionally, we have specified how active learning methodologies, such as case-based and simulation-based learning, could be practically integrated into future training to enhance the development of applied skills. We have also included a discussion on potential logistical and systemic challenges, such as funding, time constraints, and accessibility issues, that may affect nationwide implementation. While this study focused on improving pharmacists’ knowledge as a foundational step toward implementation readiness, we have acknowledged in the limitations that we did not assess confidence or counselling skills directly. We have clarified in the conclusions section that future research should explore how knowledge improvement translates into real-world counselling skills, confidence, and effective PrEP service delivery outcomes.

Response to Reviewer 2’s Comments:

Congratulations to the authors for the well written manuscript. The manuscript highlights a salient issue on the development and delivery of online PrEP training for new cadres such as pharmacy providers in the context of task shifting services. Two issues for consideration:

Comment 1: The title of the manuscript gives prominence to the development and delivery of the online training. However, the primary substance of the manuscript is on P

---

## [Decision Letter · Decision Letter 1]

21 May 2025

PONE-D-24-57343R1Development and evaluation of an online HIV pre-exposure prophylaxis (PrEP) training program for community pharmacists to implement pharmacy-led PrEP services in Malaysia.PLOS ONE

Dear Dr. Zaki,

Thank you for submitting your manuscript to PLOS ONE. After careful consideration, we feel that it has merit but does not fully meet PLOS ONE’s publication criteria as it currently stands. Therefore, we invite you to submit a revised version of the manuscript that addresses the points raised during the review process.

We look forward to receiving your revised manuscript.

Kind regards,

Ali Ahmed, PhD

Academic Editor

PLOS ONE

Journal Requirements:

Reviewers' comments:

Reviewer's Responses to Questions

**Comments to the Author**

1. If the authors have adequately addressed your comments raised in a previous round of review and you feel that this manuscript is now acceptable for publication, you may indicate that here to bypass the “Comments to the Author” section, enter your conflict of interest statement in the “Confidential to Editor” section, and submit your "Accept" recommendation.

Reviewer #4: (No Response)

2. Is the manuscript technically sound, and do the data support the conclusions?

Reviewer #4: Yes

3. Has the statistical analysis been performed appropriately and rigorously? 

Reviewer #4: Yes

4. Have the authors made all data underlying the findings in their manuscript fully available?

Reviewer #4: Yes

5. Is the manuscript presented in an intelligible fashion and written in standard English?

Reviewer #4: Yes

6. Review Comments to the Author

Reviewer #4: My comments are:

Line 106: Can you explain what you mean by strategic task shifting?

What modifications were made while adapting the physician-targeted training program for pharmacists?

Line 402: The word ‘cisgender’ may need explanation.

Can more details about the in-person workshop be provided?

Lines 486 to 489. A sentence is repeated.

7. PLOS authors have the option to publish the peer review history of their article (what does this mean? ). If published, this will include your full peer review and any attached files.

**Do you want your identity to be public for this peer review?** For information about this choice, including consent withdrawal, please see our Privacy Policy .

Reviewer #4: **Yes: ** Pathiyil Ravi Shankar

---

## [Author Response · Author response to Decision Letter 2]

3 Jul 2025

Dr. Rafdzah Ahmad Zaki

Public Health Medicine Specialist and Professor,

Department of Social and Preventive Medicine,

Faculty of Medicine

Universiti Malaya, Malaysia

rafdzah@ummc.edu.my

25th June 2025

Editor-in-Chief

PLOS ONE

Subject: Rebuttal Letter for Manuscript Revision [PONE-D-24-57343R1] - [EMID:7ee32a2b256e6417]

Dear Dr. Ali Ahmed,

We sincerely appreciate the opportunity to revise our manuscript, titled "Development and evaluation of an online HIV pre-exposure prophylaxis (PrEP) training program for community pharmacists to implement pharmacy-led PrEP services in Malaysia." (Manuscript ID: PONE-D-24-57343R1).

We are grateful for the constructive feedback provided by the reviewer. We have carefully addressed each of the comment and have made the necessary revisions to enhance the clarity and quality of our manuscript. Below, we provide a point-by-point response to each comment, detailing the changes made.

Comment: Please review your reference list to ensure that it is complete and correct. If you have cited papers that have been retracted, please include the rationale for doing so in the manuscript text or remove these references and replace them with relevant current references. Any changes to the reference list should be mentioned in the rebuttal letter that accompanies your revised manuscript. If you need to cite a retracted article, indicate the article’s retracted status in the References list and also include a citation and full reference for the retraction notice.

Response: We have carefully reviewed the reference list to ensure its completeness and accuracy. All citations were cross-checked, and no retracted articles were identified. Therefore, no rationale for including retracted references was necessary. Minor formatting corrections were also made where appropriate, and these are reflected in the revised manuscript.

The reference below has been added to replace the previously cited newspaper articles, as the official statistics on the number of PrEP users as well as the number of government clinics providing PrEP services are now available in the Malaysian Global AIDS Monitoring Report 2024.

3. Ministry of Health Malaysia. 2024 Global AIDS Monitoring: Country Progress Report - Malaysia. Putrajaya; Ministry of Health Malaysia; 2024.

3. Ministry of Health Malaysia. 2023 Global AIDS Monitoring: Country Progress Report - Malaysia. Putrajaya: Ministry of Health Malaysia; 2023.

8. Vethasalam R, Tan T, Gimino G, Lee B. More than 2000 individuals given PrEP to reduce HIV infections, says Dr Zaliha. Star [Internet]. 2023 Nov 7 [cited 2024 Mar 22]; Available from: https://www.thestar.com.my/news/nation/2023/11/07/more-than-2000-individuals-given-prep-to-reduce-hiv-infections-says-dr-zaliha

11. Ibrahim J, Rahim R, Lai A, Pfordten D. Ten more govt clinics to start dispensing HIV prevention drugs, says Dr Dzulkefly. Star [Internet]. 2024 Mar 20 [cited 2024 Mar 22]; Available from: https://www.thestar.com.my/news/nation/2024/03/20/ten-more-govt-clinics-to-start-dispensing-hiv-prevention-drugs-says-dr-dzulkefly

Additionally, please find below the references that were added in the first revision but were not mentioned in the previous rebuttal letter.

31. McKeirnan K, Czapinski J, Bertsch T, Buchman C, Akers J. Training Student Pharmacists to Perform Point-of-Care Testing. American Journal of Pharmaceutical Education. 2019;83(7):7031.

32. Alzahrani MM, Alghamdi AA, Alghamdi SA, Alotaibi RK. Knowledge and Attitude of Dentists Towards Obstructive Sleep Apnea. International Dental Journal. 2022;72(3):315–21.

41. Loh P, Chua SS, Karuppannan M. The extent and barriers in providing pharmaceutical care services by community pharmacists in Malaysia: a cross-sectional study. BMC Health Services Research. 2021;21:1–14.

Response to Reviewer 4’s Comments:

Comment 1: Line 106: Can you explain what you mean by strategic task shifting?

Response: We have revised the sentence to clarify the meaning of strategic task-shifting as follows:

"Furthermore, community pharmacists are well-trained healthcare professionals and well-positioned to provide PrEP services through strategic task-shifting, which involves the appropriate reallocation of responsibilities such as counselling and dispensing from physicians to pharmacists to improve service delivery and access."

Comment 2: What modifications were made while adapting the physician-targeted training program for pharmacists?

Response: We have clarified this in the revised manuscript under “training program development”. While adapting the physician-targeted HIV Connect PrEP course for pharmacists, we simplified certain content to focus more on oral PrEP (tenofovir disoproxil fumarate/emtricitabine), rather than options that were either unavailable or not approved for PrEP in Malaysia at the time, such as tenofovir alafenamide/emtricitabine (TAF/FTC), long-acting injectable cabotegravir, and the dapivirine ring. A summary of special circumstances was added to aid comprehension. Additionally, implementation-focused videos were incorporated to provide practical guidance on key aspects of the pharmacy-led PrEP service, including checklists for PrEP initiation and follow-up, adherence assessment and counselling, and referral pathways (when and who to refer). These modifications were made to align the training with pharmacists’ scope of practice and enhance implementation readiness.

Comment 3: Line 402: The word ‘cisgender’ may need explanation.

Response: We appreciate the reviewer’s comment. “Cisgender” is a commonly used term in public health literature and reflects the terminology used in the cited article. However, to improve clarity and consistency, we have revised the term to “cisgender men who have sex with men (MSM)” in the manuscript.

Comment 4: Can more details about the in-person workshop be provided?

Response: Thank you for this suggestion. As the primary focus of this manuscript is the development and evaluation of the online PrEP training program, we have kept the description of the in-person workshop concise in the Materials and methods section to avoid shifting focus. However, additional details about the workshop have been included in the Discussion section (lines 481 to 494; and lines 544 to 546) to provide further context on its role as a supplementary reinforcement activity.

Comment 5: Lines 486 to 489. A sentence is repeated.

Response: The repeated sentence has been removed.

We hope that our revisions adequately address the concerns raised. Please let us know if further modifications are required. Once again, we appreciate the reviewer’s and editor’s valuable feedback, which has helped us improve our manuscript.

Best regards,

Dr. Rafdzah Ahmad Zaki

Public Health Medicine Specialist and Professor,

Department of Social and Preventive Medicine,

Faculty of Medicine

Universiti Malaya, Malaysia

rafdzah@ummc.edu.my

---

## [Editor Report · Decision Letter 2]

6 Jul 2025

Development and evaluation of an online HIV pre-exposure prophylaxis (PrEP) training program for community pharmacists to implement pharmacy-led PrEP services in Malaysia.

PONE-D-24-57343R2

Dear Dr. Rafdzah,

We’re pleased to inform you that your manuscript has been judged scientifically suitable for publication and will be formally accepted for publication once it meets all outstanding technical requirements.

Kind regards,

Ali Ahmed, PhD

Academic Editor

PLOS ONE

---

## [Editor Report · Acceptance letter]

PONE-D-24-57343R2

PLOS ONE

Dear Dr. Zaki,

I'm pleased to inform you that your manuscript has been deemed suitable for publication in PLOS ONE. Congratulations! Your manuscript is now being handed over to our production team.

Kind regards,

on behalf of

Dr. Ali Ahmed

Academic Editor

PLOS ONE